# Contribution of Village Chickens in Sustainable and Healthy Food Systems for Children along a Rural–Urban Gradient: A Systematic Review

**DOI:** 10.3390/foods12193553

**Published:** 2023-09-25

**Authors:** Cresswell Mseleku, Michael Chimonyo, Rob Slotow, Lindokuhle Christopher Mhlongo, Mjabuliseni S. C. Ngidi

**Affiliations:** 1School of Agricultural, Earth and Environmental Sciences, University of KwaZulu-Natal, Pietermaritzburg 3209, South Africa; 212504154@stu.ukzn.ac.za (C.M.); 213520360@stu.ukzn.ac.za (L.C.M.); 2Faculty of Science, Engineering and Agriculture, University of Venda, Thohoyandou 0950, South Africa; michael.chimonyo@univen.ac.za; 3School of Life Sciences, University of KwaZulu-Natal, Pietermaritzburg 3209, South Africa; slotow@ukzn.ac.za

**Keywords:** anthropometry, consumption patterns, dietary diversity, flock sizes, indigenous poultry ownership

## Abstract

Achieving sustainable and healthy food systems in support of human and planetary health is a global challenge. We systematically reviewed papers (*n* = 2322) showing how village chicken products (meat, eggs, and offal) contribute to sustainable and healthy food systems for children along a rural–urban gradient. A total of 72 articles, representing all different sections covered in this review, were finally included. Production of village chickens contributed positively on livestock-derived food consumption by children. Households who owned chickens were likely to move from medium to high dietary diversity. Children from households that owned chickens had lower odds of anemia (1.07) and higher height-for-age (HAZ) and weight-for-age (WAZ) scores. Egg intervention increased HAZ and WAZ for children by 0.63 [95% confidence interval (CI), 0.38–0.88] and 0.61 [95% CI, 0.45–0.77], respectively. Village chicken ownership had positive effect on children’s poultry meat and egg consumption. Per capita consumption of chicken by girls and boys combined was 38.2 g/day, where girls had 36.9 g/day and boys had 39.4 g/day. Children from households that owned chickens consumed eggs 2.8 more times per week compared to children from households without chickens. Moving along a rural–urban gradient, village chicken production was less common. Improved production and quality of village chicken products, and policies supporting optimal maternal and child intake these products are required.

## 1. Introduction

A food system encompasses food production, processing and packaging, distribution and retailing, and consumption [1]. Achieving sustainable and healthy food systems in support of human and planetary health is a global challenge [2]. Currently, malnutrition is mostly prevalent in sub-Saharan Africa, with an approximation of 50% of the world’s undernourished human population inhabiting this region [3]. Child undernutrition remains a global challenge, with over 20% of children (<five years of age) being stunted and 7.5% suffered wasting in 2017. The profitability of village chicken production and its potential of combating hunger and malnutrition has been globally recognized by many scholars and policy makers. Village chicken production systems have been integrated with the livelihoods of resource-poor households for improved income, diet, and food and nutrition security [4].

The importance of village chicken production and consumption has been steadily increasing in the developing countries due to its contribution to the income generation and nutritional status of smallholder farmers [5]. Production of village chickens is less detrimental to the environment compared to other livestock production [5]. Village chicken products (meat, eggs, and offal) form an integral part in human diets as they provide essential nutrients that are barely obtained from plant-derived products [6]. Village chicken products contain high-quality protein and macro- and micro-nutrients [7]. Inadequate micro-nutrient intakes are associated with nutritionally related health issues, especially for children [8]. Most common health issues include anemia, stunting, and low birthweights [9,10]. Village chicken products are underutilized; therefore, their utilization could potentially improve nutrient intake and further combat nutritional challenges in children.

Children during complementary feeding phase (6–59 months of age) are vulnerable to nutritionally related health issues [11]. Ensuring proper complementary feeding phase for children that meets their nutritional requirements is crucial [12]. Village chicken products are, however, neglected regardless of their nutrient density which could play an important role in children’s diet during complementary feeding phase. There is, therefore, a need to utilize these neglected food items to meet nutritional requirements for children. Village chickens are gaining more preference than exotic breeds due to their lean meat [13]. The importance of village chickens combined with consumers’ preference for their products indicate that these genetic resources are potential options for combating food and nutrition insecurity. The growing demand for village chicken products across rural and urban areas is an opportunity for the enhancement of village chicken production in Africa [5]. Village chickens are mostly reared by smallholder farmers exclusively in rural and peri-urban areas, and there is little information available regarding their production in urban areas. There is paucity of knowledge regarding the changes in the contribution of village chickens to sustainable and healthy food systems along the rural–urban gradient. Using a rural–urban gradient could, therefore, reveal the contribution of village chicken products between the urban center and surroundings.

The broad objective of the study was, therefore, to assess the contribution of village chicken products in sustainable and healthy food systems for children along a rural–urban gradient. The specific objectives were to determine: (1) the consumption patterns of village chicken products along a rural–urban gradient (2) the relationship between village chicken ownership and diet diversity of households along a rural–urban gradient (3) the relationship between the production of village chicken products with consumption patterns and nutrient intake along a rural–urban gradient; and (4) the relationship between consumption patterns of village chicken products with child welfare and development along a rural–urban gradient.

## 2. Materials and Methods

A systematic review on the contribution of village chicken products in sustainable and healthy food systems for children along a rural–urban gradient was performed following the Preferred Reporting Items for Systematic Reviews and Meta-Analyses (PRISMA) guidelines (www.prisma-statement.org, accessed on 1 October 2022).

### 2.1. Eligibility Criteria

The eligibility of a study was assessed in terms of the population investigated, intervention types, comparisons undertaken, outcome variables, and research designs. For eligible populations, studies included were on women and children. Studies accepted were those that: (i) compared farmers or communities, and where village chicken products were consumed; (ii) were accessible for full review and written in English; (iii) were reporting original research (not literature review); (iv) were focusing on village chicken products (meat, eggs, and offal; livers, intestines, gizzards, hearts, and head and feet); and (vi) were published in a peer reviewed journal (not pre-prints). Separate criteria were used to determine the eligibility for quantitative versus qualitative synthesis. For the quantitative synthesis, accepted studies were those that; (1) used either a randomized experiment; and (2) used a quasi-experimental design. For the qualitative synthesis, studies that were accepted are: (1) studies that had a clearly defined research objectives, and links to the relevant literature; and (2) studies that provided details on context, sample selection, and data collection methods.

### 2.2. Source of Information and Search Strategy

The literature for the review was drawn from scientific journal articles by searching Google scholar, Science direct, Scopus, and Web of Science. The search was limited to papers from 2010 to the present. All main search themes were conducted separately in respective order (Table 1).

### 2.3. Selection Process

Papers published in any language other than English, not peer-reviewed, or which presented only an abstract were excluded. The literature used was regardless of the research method employed the authors (qualitative and or quantitative). Inter-rater reliability was used, and screening was performed independently by two researchers to select papers based on the relevance of the title and abstract to the topics listed in the objectives. Discrepancies were resolved through face-to-face discussions by the two researchers. Articles identified from title and abstract screening were reviewed in full texts. The two researchers jointly determined the final articles to be included in the review.

### 2.4. Risk of Bias Assessment

Risk of bias for this study was independently assessed by C.M. and L.C.M. for all included studies using the Joanna Briggs Inventory (JBI) checklist [14]. Disagreements were resolved by M.S.C.N. The following nine criteria were used in the assessment of study risk of bias: (1) Was the sample frame suitable to address the targeted population? (2) Were the participants of the study appropriately? (3) Was the sample size proper? (4) Were the study subjects and settings detailed? (5) Was data analysis conducted with sufficient coverage of the identified sample? (6) Were reliable methods used for the identification of the condition? (7) Was the condition measured in a standard, reliable way for all participants? (8) Was there appropriate statistical analysis? (9) Was the response rate adequate, and if not, was the low response rate managed appropriately? Assessment of the level of bias was carried out by calculating the total number of criteria with a ‘yes’ responses and converting the scores into percentages (n/9). Studies that scored less than 50% were expressed as high risk of bias, 50–69% was medium risk of bias, and ≥70% was low risk of bias.

### 2.5. Quality Assessment

The quality of eligible studies was assessed according to Law et al. [15] criteria. Following these criteria; purpose/objective, the literature background, design, sample/population, intervention, results, implications, and conclusions were assessed. Each of these components was given a score of 1 (meets the criterion), or 0 (does not meet the criterion). A maximum score of 15 items was used as the measure of the quality of studies. Only studies given a score of 9 or above were included.

## 3. Results

### 3.1. Characteristics of Included Studies

The search from all databases produced 2322 records. After the removal of duplicates (*n* = 728), the remaining records (*n* = 1594) were screened for eligibility. Of these articles, 1516 articles were excluded in the first round and 76 remained. In the second round of screening full-texts, six further articles were excluded. A total of 72 articles, representing all different sections covered in this review, were included (Figure 1).

### 3.2. Production of Village Chickens

Village chicken production can be referred to not only as the production of meat and eggs, but also as the distribution and selling of these products [16]. Production of village chickens is extensively based, being characterized by low productivity and poor management [17]. Village chicken production benefited rural households in terms of less rearing costs and management practices [18]. In households from the peri-urban area, village chicken production contributed to food security through income derived from chicken sales [19].

Production of village chickens at a household level is one of the sensitive nutritional approaches to improve consumption of meat and eggs and nutrition status [20]. In rural and urban areas of Kenya, production of village chicken meat was identified as a necessity [21]. In Africa, village chicken production was mainly challenged by disease outbreak (Newcastle), predators, poor nutrition (quality and quantity of feed material), poor marketing system, and poor genetic potential [5]. Incorporating improvements in management practices, nutrition, genetics, and disease control could potentially help in achieving profitable and sustainable village chicken production that benefits smallholder farmers [22]. Production of village chickens contributed positively on the livestock-derived food consumption [20]. Moving along a rural–urban gradient, production of village chickens decreased, with more households rearing them in rural areas compared to peri-urban and urban areas [23].

### 3.3. Village Chickens and Their Sustainability

Village chicken sustainability depends on their production input and consumption [24]. For sustainable village chicken production, it is, however, imperative to consider other elements along the value chain, rather than focusing on the two ends of value chain which are production and consumption [25]. Understanding flock dynamics was a key to sustainability of village chicken production [26]. Bettridge et al. [24] stated that, sustainable village chicken development interventions for small-scale farmers, including breeding programs, should be tailored locally, and designed for adaptable implementation. Sustainability of chicken meat was mostly associated with the environmental impact of its production, followed by animal welfare dimensions [27]. Garcez de Oliveira Padilha et al. [27] revealed that high chicken meat price was associated with high sustainability of chickens.

### 3.4. Nutrient Profiles of Meat, Eggs, and Offal

Spearing et al. [28] found that fried chicken meat contained 1.47 MJ, 29.7 g, and 23.7 g of energy, protein, and fat per 100 g, respectively. Protein is required for fundamental human health and growth, and its requirements can be met through adequate intakes of village chicken meat, eggs, and offal [29]. Chicken breasts were richer in protein with low fat content (Table 2). In Accra region, Ghana, household mean protein intake per month from village chickens was 0.11 kg [30].

#### Nutrient Bioavailability

Village chicken products contained proteins with a digestibility of up to 98% in a readily available form, whereas plant food sources contained 75–85% [4]. In low- and middle-income countries (LMICs), intake of iron, iodine, vitamin A, and zinc food sources was irregular [36]. Hence, poor intake of these micro-nutrients, however, resulted to anemia, diarrhea, impaired cognitive function, and childhood blindness [37]. High contents of micronutrients in chicken products are important for physical growth, cognitive function, and performance [38]. Spearing et al. [28] found that 100 g of fried chicken contained 1.7 mg, 3.1 mg, and 47.0 µg-RAE of iron, zinc, and vitamin A, respectively. Chicken meat contained higher amounts of vitamin A, which is equivalent to those of beef and pork [5]. Chicken liver was dense in micro-nutrients, specifically iron and vitamin A (Table 3). In Zambia, Asamane et al. [39] found that intakes of micro-nutrients (zinc, iron, vitamin A, vitamin B6, and vitamin B12) from meat were higher in an urban area (Lusaka) than in a rural area (Chongwe).

### 3.5. Chicken Ownership and Dietary Diversity for Children

Dietary diversity is a summed number of different food groups consumed at a given time, and it is referred as a proxy to food security [44]. The diversity of diets for children mainly depends on crops and livestock diversification and nutrition knowledge that caregivers possess. Nutrition knowledge, however, commonly improves children’s diet diversity only from areas with good access to the markets [45]. Ochieng et al. [46] reported that dietary diversity is a good indicator of dietary quality, and it is positively associated with the nutritional status. In the Eastern Cape Province, South Africa, village chickens were the most kept livestock species, with over 79% and 81% of chickens in Centane and Mount Frere were mainly owned by women [17]. Village chicken ownership contributed directly and indirectly to child’s food and nutrition security. Direct contribution was improved household diet through access to chicken meat, whereas indirect contribution was chickens being sold to purchase diverse foods [3].

Chicken ownership was positively associated with increased dietary diversity, and households who owned chickens were more likely to move from medium to high dietary diversity [47]. In Halaba, Ethiopia, Yemane et al. [48] found that the average village chicken flock size was 8.5 chickens. Gitungwa et al. [49] found that owning large village chicken flock increased the probability of food security in pastoralist households; however, it decreased it in agropastoralist households, while increasing the likelihood of medium-high dietary diversity. Dietary diversity was influenced by gender and educational level of household heads and agricultural land size [46]. Female-headed households and households with educated heads had more diverse diets [46,49]. Dumas et al. [50] reported that having a small-sized chicken flock was associated with decreased child dietary diversity and increasing the flock size was positively associated with improved dietary diversity. Households who owned large number of village chickens had greater diversity of consumed animal-sourced foods among children [51]. In Kenya, Makueni District, village chicken flock size was positively associated with household income generation [52]. Increase in flock size was associated with increased child’s dietary diversity (Table 4). Pathways from household village chicken ownership to improved household welfare and resilience (Figure 2).

### 3.6. Chicken Ownership, and Child Welfare and Development

Children from households who owned chickens had lower odds of anemia (1.07) compared to those from households who owned no livestock at all [55]. Traditional ownership of village chickens, however, potentially presented health and nutrition risks due to poultry scavenging on compounds that may increase children’s exposure to livestock-related pathogens [56]. De Bruyn et al. [57] found no association between chicken flock size and child diarrhea. Large flock size contributed to the presence of too much chicken feces which was associated with lower weight-for-height (WHZ) scores, and there were no associations found with height-for-age (HAZ) scores [56]. Hetherington et al. [3] reported that in Bonsaaso, Ghana and Pampaida, Nigeria, children from households who owned chickens had higher (better) anthropometric measurements (HAZ and weight-for-age; WAZ scores). Village chicken ownership was uncommon when moving along the rural–urban gradient due to high cost of feed [58].

### 3.7. Consumption Patterns of Village Chicken Products

In Lesotho, chicken meat was frequently consumed in urban than in rural areas [59]. In the Chris Hani Municipality, Eastern Cape Province of South Africa, village chicken meat consumption was higher in warmer seasons than in cold seasons [60]. In Ga East and Shai Osudoku Districts, Ghana, chicken meat was consumed mostly by children, on average, approximately once per week [55]. Bundala et al. [20] reported that the findings from focus group discussions revealed that small number of chickens, lack of nutrition knowledge, and monetary and cultural values attached to chickens hampered meat and offal consumption. In rural areas, increased farm size, literacy, and income, and better knowledge on health and nutrition should improve per capita consumption of chicken meat [61].

Ndenga et al. [62] found that consumption frequency of village chicken meat was affected by age, gender, education, and household size. Hetherington et al. [3] found that in Bonsaaso, Ghana and Pampaida, Nigeria, children who consumed chicken meat in the last 30 days had higher HAZ, WAZ, and weight-for-height (WHZ). Iannotti et al. [43] found that egg intervention increased HAZ and WAZ for children by 0.63 [95% confidence interval (CI), 0.38–0.88] and 0.61 [95% CI, 0.45–0.77], respectively. In Mkhambathini municipality, KwaZulu-Natal, South Africa, Kolanisi et al. [63] found that 58% of households who owned village chickens consumed their meat once a month. Ownership and flock size of village chickens had positive effect on poultry meat consumption [64]. In Rwanda (Mayange) and Senegal (Ruhiira), village chicken meat consumption was higher in poultry-keeping households. In Zambia, chicken was the most frequently consumed meat compared to other meat types in both rural and urban areas [39]. Sui et al. [65] found that per capita consumption of chicken meat for children (girls and boys combined) was 38.2 g/day, where girls had 36.9 g/day and boys had 39.4 g/day. Children from households that owned one or more chickens consumed eggs 2.8 more times per week compared to children from households that had no chickens [66].

### 3.8. Dietary Transition: Threats and Opportunities

Delisle et al. [67] revealed a positive rural–urban gradient in nutrition transition, poultry meat consumption increased along with movement from rural to urban areas. Dietary transitions from meat-based to a more plant-based diet requires establishment of convenient meat substitutes [68]. Although plant-based diets are not yet found to be interfering with human health and are promising in sustaining the environment and animal welfare, animal-sourced foods are, however, dense in essential nutrients. Transitions from meat-based to plant-based diets threatens the acquisition of sufficient nutrients found in animal-sourced foods; however, village chicken meat is ideal for consumption due to its low levels of cholesterol and nutrient density.

## 4. Discussion

The present study examined research articles from 2010 to the present, based on the information obtained from Google scholar, Science direct, Scopus, and Web of science. We reviewed scientific evidence of the contribution of village chicken products to sustainable and healthy food systems for children along a rural–urban gradient. The literature on production and consumption of village chicken offal is scanty, regardless of their nutritional importance ascribed to high contents of macro- and micro-nutrients. This, however, presents an opportunity for researchers to research on these aspects. Low contribution of village chicken meat to sustainable and healthy food systems could be attributed to their small flock sizes [30]. Households that own few village chickens are unlikely to slaughter or sell, due to the economic incentive of allowing the flock to grow first in order to capitalize on the investment of buying those chickens [69]. Positive contribution of village chicken ownership to dietary diversity is attributed to several food items (meat and eggs), and easiness of rearing and trading [47]. Village chickens further contribute positively to dietary diversity through their manure used for growing crops [70]. Production of village chickens was mostly beneficial to rural households, a similar finding by Moreki et al. [71], which could be ascribed to ease of accessibility and species abundancy. The observed positive association between consumption of village chicken meat and child development could be attributed to the presence of macro- and micro-nutrients in meat that are responsible for growth and development of children [3,38]. The observed low per capita consumption of chicken meat by girls compared to boys could be due to gender prioritization, similar findings were reported by Dumas et al. [50], where it was found that women had limited access to meat. The movement along a rural–urban gradient is associated with households becoming landless, resulting to the lack of agricultural and husbandry practices [72]. The heterogeneity in the contribution of village chickens is strongly associated with area, husbandry, and flock sizes. A study by Chaiban et al. [73] observed heterogeneity on production systems of village chickens in terms of their age, size, accessibility, husbandry, opportunities, and challenges [74].

The limitations of evidence for this systematic review include search terms, potential confounders (paucity of evidence and region) that were not examined in most or any of the original research, the keywords, and possible publication bias. Search terms used limited evidence due to other study components not being captured. Paucity of evidence was due to localization of studies and there were no available large-scale multi-disciplinary studies. The quality of included studies was assessed according to Law et al. [15] criteria. Following the criteria; purpose, the literature background, design, sample, outcomes, interventions, results, implications, and conclusions were assessed. Each of these components was given a score of 1 (meets the criterion), or 0 (does not meet the criterion). A maximum score of 15 items was used as the measure of the quality of studies. Only studies given a score of 9 or above were included.

## 5. Conclusions

The potential of village chicken product’s contribution to sustainable and healthy food systems has long been recognized; however, its impact has been scantly difficult to demonstrate. Consumption of village chicken products by children remain low, more especially of offal, specifically by girls. Age, gender, lack of education, and household size were the main constraints of consumption of village chicken products. Village chicken meat, eggs, and offal were mostly beneficial to households in rural areas. Health issues such as diarrhea, and anemia prevail in young children under five years of age. Interventions that aim to improve the production and quality of village chicken products and policies supporting optimal maternal and child intake of these products are required. To improve the contribution of village chickens in a sustainable and healthy food systems, employment and income opportunities are needed in agriculture and on the off-farm activities for income generation. A research focus on the production of village chickens must be employed across the rural–urban gradient, rather than focusing on rural and peri-urban areas only. The nutritional requirements of village chickens are unknown; therefore, a research focus on developing the nutritional guidelines for village chickens is required to maximize their productivity. This study recommends that the intervention of agricultural extension institutions is highly required for the improvement of village chicken production.

## Figures and Tables

**Figure 1 foods-12-03553-f001:**
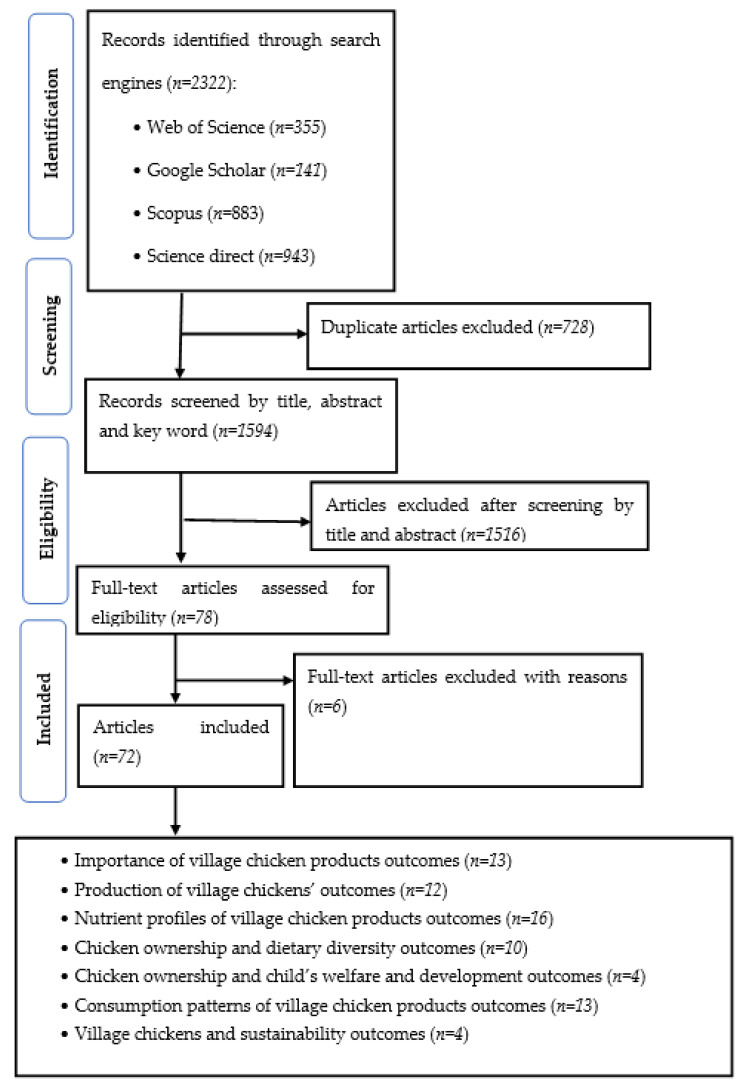
Flow diagram of selected studies.

**Figure 2 foods-12-03553-f002:**
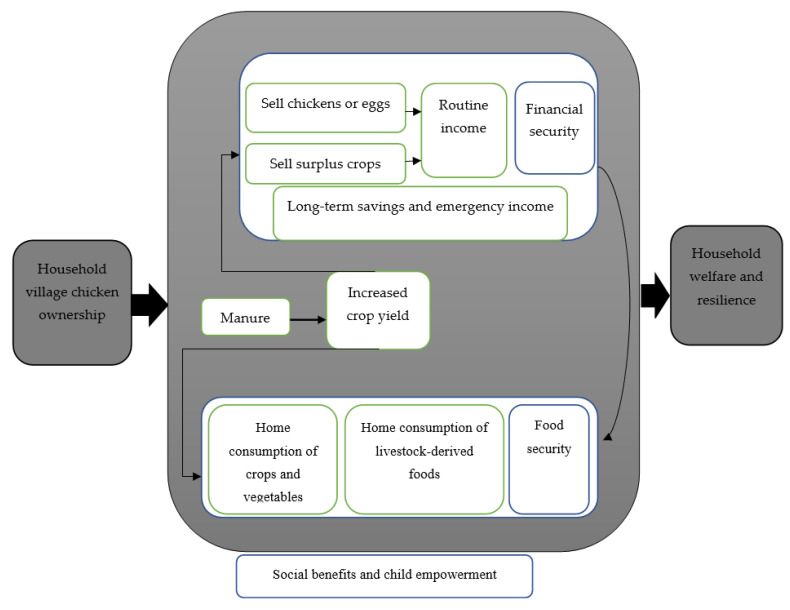
Pathways from household village chicken ownership to improved household welfare and resilience.

**Table 1 foods-12-03553-t001:** Searching order and search themes used.

1. “importance” OR “role” AND “village chicken” OR “scavenging chicken” OR “backyard chicken” AND “products” OR “meat” OR “eggs” OR “offal” OR “gizzards” OR “livers” OR “intestines” AND “sustainable food system” OR “healthy food system” AND “household” OR “community” AND “rural areas” OR “peri-urban areas” OR “urban areas” AND “Africa” OR “developing countries”.
2. “production” OR “yield” AND “village chicken” OR “scavenging chicken” OR “backyard chicken” AND “products” OR “meat” OR “eggs” OR “offal” OR “gizzards” OR “livers” OR “intestines” AND “sustainable food system” OR “healthy food system” AND “household” OR “community” AND “rural areas” OR “peri-urban areas” OR “urban areas” AND “Africa” OR “developing countries”.
3. “nutritive value” OR “nutrient profile” OR “nutrient bioavailability” AND “nutrient intake” AND “village chicken” OR “scavenging chicken” OR “backyard chicken” AND “products” OR “meat” OR “eggs” OR “offal” OR “gizzards” OR “livers” OR “intestines” AND “sustainable food system” OR “healthy food system” AND “household” OR “community” AND “rural areas” OR “peri-urban areas” OR “urban areas” AND “Africa” OR “developing countries”.
4. “utilization” OR “use” AND “village chicken” OR “scavenging chicken” OR “backyard chicken” AND “products” OR “meat” OR “eggs” OR “offal” OR “gizzards” OR “livers” OR “intestines” AND “sustainable food system” OR “healthy food system” OR “household” OR “community” OR “rural areas” OR “peri-urban areas” OR “urban areas” AND “Africa” OR “developing countries”.
5. “dietary diversity” OR “diet diversity” OR “dietary transition” AND “village chicken” OR “scavenging chicken” OR “backyard chicken” AND “products” OR “meat” OR “eggs” OR “offal” OR “gizzards” OR “livers” OR “intestines” AND “sustainable food system” OR “healthy food system” AND “household” OR “community” AND “rural areas” OR “peri-urban areas” OR “urban areas” AND “Africa” OR “developing countries”.
6. “ownership” OR “flock size” AND “village chicken” OR “scavenging chicken” OR “backyard chicken” AND “products” OR “meat” OR “eggs” OR “offal” OR “gizzards” OR “livers” OR “intestines” AND “sustainable food system” OR “healthy food system” AND “household” OR “community” AND “rural areas” OR “peri-urban areas” OR “urban areas” AND “Africa” OR “developing countries”.
7. “consumption” OR “consumption patterns” AND “village chicken” OR “scavenging chicken” OR “backyard chicken” AND “products” OR “meat” OR “eggs” OR “offal” OR “gizzards” OR “livers” OR “intestines” AND “sustainable food system” OR “healthy food system” AND “household” OR “community” AND “rural areas” OR “peri-urban areas” OR “urban areas” AND “Africa” OR “developing countries”.
8. “sustainability” AND “village chicken” OR “scavenging chicken” OR “backyard chicken” AND “products” OR “meat” OR “eggs” OR “offal” OR “gizzards” OR “livers” OR “intestines” AND “sustainable food system” OR “healthy food system” OR “household” OR “community” AND “rural areas” OR “peri-urban areas” OR “urban areas” AND “Africa” OR “developing countries”.

**Table 2 foods-12-03553-t002:** Macro-nutrient contents of village chicken products.

References	Chicken Part	Protein (g/kg)	Fat (g/kg)	Energy (MJ/kg)
[31]	Gizzard	16.1 ± 0.16	-	-
[31]	Liver	19.4 ± 0.33	-	-
[32]	Breast	24.2 ± 0.16	1.24 ± 0.09	17.83 ± 16.4
[33]	Drumstick	18.0 ± 0.96	-	-
[34]	Leg	18.5 ± 0.93	7.28 ± 1.69	-
[35]	Thigh	13.7 ± 0.04	5.64 ± 0.35	-

**Table 3 foods-12-03553-t003:** Concentrations of micro-nutrient contents in chicken meat, eggs, and offal.

Reference	Chicken Part	Iodine (mg/100 g)	Iron (mg/100 g)	Zinc (mg/100 g)	Vitamin A (µg/100 g)
^1^ [40]; ^4^ [41]	Chicken (muscle)	^4^ 0.003–0.096	^1^ 5.5	^1^ 10.0	^1^ 3400.0
^2^ [31]	Gizzard	-	^2^ 18.1	^2^ 18.6	-
^2^ [31]; ^4^ [41]; ^5^ [2]	Liver	^4^ 0.008–0.225	^2^ 80.9	^2^ 26.1	^5^ 9700.0
^3^ [42]; ^5^ [2]	Breast	-	^5^ 0.7	^3^ 0.8	^5^ 27.0
^6^ [43]	Egg	-	^6^ 2.0	^6^ 1.1	^6^ 106.9

The numbers show that the values in table corresponds with which reference (source).

**Table 4 foods-12-03553-t004:** Relationship between village chicken flock sizes and child’s dietary diversity scores ((mean ± standard deviation) (regression coefficient)).

References	Flock Sizes (n)	Dietary Diversity Score
[53]	311	4.0 ± 1.8
[54]	93.4	2.6 ± 1.1

## Data Availability

Data is contained within the article.

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
