# Peer review of "Contribution of Village Chickens in Sustainable and Healthy Food Systems for Children along a Rural–Urban Gradient: A Systematic Review"

_foods, 2023, doi:10.3390/foods12193553_

Round 1
Reviewer 1 Report
A systematic review paper includes relevant studies, completeness of synthesis, and clarity and transparency of presentation. The paper formulates clear research objectives and questions that relate directly to the topic of interest. The research question, which examines how village chicken products (meat, eggs and offal) contribute to sustainable and healthy food systems for children along an urban- rural divide, is clearly defined.
The systematic review adequately assessed the methodological quality of the included studies. The use of standardised quality assessment tools adds credibility to the findings.
The paper is well structured, with clear headings and subheadings so that the reader can easily find their way around. The reporting of results is transparent and includes detailed descriptions of the included studies.
Areas for improvement:
The authors have conducted a thorough and comprehensive search of multiple databases, reducing the likelihood of overlooking relevant studies. The inclusion and exclusion criteria are well justified, but there is a lack of definition of the timing of the selected analysis. Why were studies from 2022 not included?
While the review acknowledges the possibility of publication bias, a more detailed investigation of this issue would have been useful. The use of funnel plots or other methods to assess publication bias should be considered.
The review acknowledges heterogeneity among the included studies, but does not elaborate on its causes or potential impact on the results. Insight into the sources of heterogeneity would increase the robustness of the study.
Although the review presents a clear synthesis of the evidence, it could benefit from a more detailed discussion of the implications of the findings. How can these findings be applied in a real-world context?
Reviewer 2 Report
Dear authors,
The first step in conducting a systematic review is to identify the research question. This needs to be specified very clearly. Authors should clearly formulate the research question in the introductory part of the manuscript so that it is unambiguous and well structured.
The authors used the following eligibility criteria: "The eligibility of a study was assessed in terms of the population investigated, in-86 tervention types, comparisons undertaken, the outcomes variables, and the research de-87 sign. For eligible populations, studies included were on women and children. Studies 88 accepted were those that: (i) compared farmers or communities, and where village 89 chicken products were consumed; (ii) were accessible for full review and written in Eng-90 lish; (iii) were reporting original research (not literature review); (iv) were focusing on 91 village chicken products (meat, eggs, and offal; livers, intestines, gizzards, hearts, and 92 head and feet); and (vi) were published in a peer reviewed journal (not pre-prints). "
Was there also a criterion related to the research area, the geographical area taken into study? Did the authors want to identify only studies from Africa or from other parts of the world?
The authors should also include some additional quality criteria on the selection of relevant studies. These could be organised in tabular form.
The results are nicely presented, but in the end it does not emerge what would be the future directions of research following the analyses carried out and their interpretation. Also, the authors should mention what is the gap identified in the literature in order to propose new research directions. What would be the managerial, social implications of these proposals for future studies?
Reviewer 3 Report
This is a very interesting review that appears to have been well conducted. The justification is clear although one issue remains for me.
Please detail how the urban-rural identification was conducted in more detail and provide further justification for this approach.
the paper is generally well written
Round 2
Reviewer 2 Report
Dear authors,
The paper was improved according to the reviewer comments.
Congratulations for your work.
Author Response
Thank you very much.